# CHATGPT LIKE TOOLS FOR GENE NETWORKS RECONSTRUCTION

## ABSTRACT

Bioinformatics tools presenting Large Language Models (LLM) models for users analyzing data specific to the knowledge domains. GPT like tools ("Generative Pre-trained Transformer") become popular in various research domains for data analysis and knowledge generation. Bioinformatics and systems biology area have series of such AI tools for gene interaction prediction - ExpasyGPT, STRING-chat, MEGA-GPT, GP-GPT, scGPT keeping 'GPT' wording in abbreviation. We consider features of these tools, focusing on gene and proteins network reconstruction, visualization and explainability. Each tool presents own systems biology ontology, that raises problem of data integration and unification. While most of tools are focused on specific areas (only single cell sequencing data, or phylogeny data) general scheme of the systems biology data presentation needs to be discussed and standardized. KEGG (Kyoto Encyclopedia of Genes and Genomes), PathBank, and Reactome remain comprehensive descriptions of available data, web-based applications such as STRING-DB and GeneMANIA combine power of AI based text generation, and web interface for convenient usage in biomedicine and disease modeling. Overall, we discuss current state of ChatGPT like tools allowing description complex biological phenomena in molecular interactions and challenges of correctness, reproducibility, and accessibility of the generated results.

## 1 INTRODUCTION

Large Language Models (LLMs) have transformed bioinformatics by enabling advanced analysis of biological data like proteins, genomics, and multi-omics, with recent ChatGPT-like tools focusing on natural language interfaces for complex queries. These models excel in tasks such as structure prediction, drug design, and data integration, accelerating research efficiency (Madani et al., 2023; Oniani et al., 2024).

This review highlights bioinformatics tools that present LLM models for users to present data specific to the knowledge domains: ExpasyGPT, STRING-Chat, and MEGA-GPT (Bolleman et al., 2025; Szklarczyk et al., 2023; Allard and Kumar, 2025; Stecher et al., 2025). The goal of these tools is to provide an ontology of systems biology based on data queries, which allows for the description of complex biological phenomena in protein structure and molecular interactions.

History of computer representation of gene and protein interaction network tools and databases in Russia may refer to GeneNetwork platform developed at the Institute of Cytology and Genetics SB RAS in Novosibirsk in 1990s. Note developing series of applications ANDsystems (Associative Network Design) - ANDvisio, ANDdigest tools developed by the same authors group. Sechenov University in Moscow works on AI applications in digital health including complex study of diseases, diseases and drugs interactions. Here we are focused on applications of ChatGPT like in gene network reconstruction for given diseases using database search and AI tools, following analysis of hub genes for drug targets search. ChatGPT like tools not only produce information in set of standard formats (text, PDF, image, sent of hyperlinks) but are also able generate full scientific report based on prompts (requests to the data). Case studies of gene network reconstruction for glioma, schizophrenia, Angelman syndrome are considered based on STRING-DB and GeneMANIA tools.

The context for the development of such interfaces is the challenge of data integration in the Life sciences. Since data is inherently diverse, differing in format, storage, and access methods, makes

data integration and reuse particularly challenging. To overcome this, the Swiss Institute of Bioinformatics (SIB) has contributed to building a "semantic web" of biological data, featuring public SPARQL endpoints for databases including Bgee, Cellosaurus, GlyConnect, STRING, and UniProt (of Bioinformatics RDF Group Members, 2024). SPARQL allows users to perform complex queries that go beyond simple text searches, as well as federated queries that retrieve data from multiple distributed sources simultaneously (Bolleman et al., 2025; Moretti et al., 2026). However, building SPARQL queries is often complex and requires specialized knowledge that many users lack, creating a significant technical barrier.

Advances in LLMs, such as ChatGPT, have opened up new possibilities in natural language processing. These models can be used to translate user questions directly into structured queries, allowing users to interact with data in a simple language without needing to understand complex syntax or the underlying knowledge graph structure (O'Neil et al., 2024). This has significantly simplified data interaction in bioinformatics.

## 2  RECENT CHATGPT-LIKE TOOLS IN BIOINFORMATICS

ExpasyGPT (`https://www.expasy.org/chat`) is an interface that uses an LLM with a Retrieval-Augmented Generation (RAG) architecture (Bolleman et al., 2025). This tool translates natural language questions into SPARQL queries, providing access to structured biological knowledge through supported endpoints. Thus, ExpasyGPT provides access to the combined knowledge graphs of SIB, significantly expanding the capabilities of searching through compatible databases (of Bioinformatics RDF Group Members, 2024).

Other specialized models complement these capabilities: scGPT (2024) for single-cell analysis and GP-GPT for genomic phenotypes (Zheng et al., 2025; Oniani et al., 2024). LLMs also support protein structure prediction, nucleic acid analysis, and multi-omics integration, often outperforming traditional methods in handling big data (Madani et al., 2023; Oniani et al., 2024). In genomics, they predict variant effects, gene expression, and regulatory networks; in drug discovery, they aid molecule generation and toxicity prediction (Oniani et al., 2024; Liu et al., 2025; Gangwal et al., 2024). Biomedical literature mining and functional annotation also benefit from their NLP capabilities (O'Neil et al., 2024; Madani et al., 2023).

Table 1 provides a comparison of notable ChatGPT-like tools in bioinformatics.

Table 1 Notable Models Comparison

| Model | Year | Focus Areas | Strengths |
|---|---|---|---|
| scGPT | 2024 | Cell annotation, multi-omics, gene networks | Aligns with known functional groups |
| GP-GPT | 2024 | Genetic phenotypes, genomic relations | Improved representations via fine-tuning |
| ExpasyGPT | 2024+ | Database querying (UniProt, etc.) | Natural language to SPARQL, federated queries |

STRING-chat (`https://string-db.org/cgi/chat`) is a natural-language interface that is built on a Model Context Protocol (MCP) server allowing users to ask questions about proteins, their functions, interactions, and related diseases (Szklarczyk et al., 2023; str, b;a). The tool converts these queries, calls multiple MCP tools sequentially, aggregates their outputs, and summarizes the combined results into a single response, often with a network visualization.

MEGA-GPT (`https://chatgpt.com/g/g-RmeN18Ssp-mega-gpt`) is an artificial intelligence-based resource that uses ChatGPT, augmented with RAG, to guide users through MEGA's analytical workflows using natural language queries (Allard and Kumar, 2025).

The tools reviewed — ExpasyGPT, STRING-chat, and MEGA-chat — demonstrate the use of LLMs as natural language interfaces for bioinformatics resources (Bolleman et al., 2025; Szklarczyk et al., 2023; Allard and Kumar, 2025; Stecher et al., 2025). Their primary goal is to bridge the gap between

complex computational data structures and the intuitive question-driven workflow of natural science researchers. By transforming natural language into precise queries and efficient workflows, they simplify access to relevant databases and tools, accelerating search and reducing technical barriers to entry.

## 3 TOOLS FOR DRUG DISCOVERY

LLMs are increasingly used in drug discovery for tasks like target identification, molecule generation, ADMET prediction, and efficacy forecasting (Oniani et al., 2024; Liu et al., 2025; Gangwal et al., 2024; Zheng et al., 2025). Recent models leverage specialized training on biological and chemical data to enhance accuracy in these areas.

Geneformer, pretrained on 30 million single-cell transcriptomes, models diseases like cardiomyopathy and identifies therapeutic targets via in silico gene deletion (Oniani et al., 2024). scKAN integrates single-cell data for cell-type-specific drug targets, outperforming differential expression methods (Zheng et al., 2025). Protein LLMs like ESM analyze sequences for evolutionary conservation and binding sites to validate targets (Madani et al., 2023).

MolGPT generates novel drug candidates with desired properties from SMILES data (Oniani et al., 2024). Specialized LLMs like those based on ESM2 enable de novo molecule design and lead optimization by predicting reactions and editing structures (Madani et al., 2023). Chemcrow and similar tools automate chemistry experiments for synthesis planning (Oniani et al., 2024).

ADMET (Absorption, Distribution, Metabolism, Excretion, and Toxicity) and Efficacy Prediction. ESM2 and Galactica predict absorption, distribution, metabolism, excretion, and toxicity (ADMET) properties, supporting zero-shot applications (Madani et al., 2023). scGPT enhances cell line drug response (CDR) prediction using single-cell embeddings, improving IC50 accuracy and generalization to unseen drugs (Zheng et al., 2025; Oniani et al., 2024). DrugGPT excels in drug interaction analysis and recommendations (Oniani et al., 2024).

Table 2 summarizes key models in drug discovery applications.

Table 2 Models Comparison in Drug Discovery

| Model | Primary Use | Key Strength |
|---|---|---|
| Geneformer | Target ID via transcriptomics | Handles sparse data for rare diseases |
| scGPT | Drug response prediction | Zero-shot cell clustering, generalization |
| ESM2/ESMFold | ADMET, structure prediction | Single-sequence folding accuracy |
| MolGPT | Molecule generation | Property-guided de novo design |

In addition to drug development, LLMs are also used to understand the mechanisms of disease development and their treatment at the molecular level. Studies have examined regulatory interactions between transcription factors and target genes using popular LLMs and control datasets (Noel et al., 2026), predicting drug-target interactions using multimodal molecular language models (Yu et al., 2026), and the assessment of LLM knowledge for rare diseases (Groza et al., 2026).

Protein abundance data across organisms has been curated and analyzed in PaxDb v6.0, providing valuable quantitative resources for understanding protein expression patterns and their implications for drug targeting (Huang et al., 2026).

LLMs are also transforming access to biological databases, as demonstrated by the Swiss Institute of Bioinformatics resources discussed below. These databases, including UniProt, Bgee, and STRING, provide the foundational data that powers both traditional bioinformatics analysis and emerging LLM-based approaches.

## 4 SWISS INSTITUTE OF BIOINFORMATICS RESOURCES AND SPARQL ENDPOINTS

The SIB (Swiss Institute of Bioinformatics) has been developing high-quality databases to serve the scientific community (of Bioinformatics RDF Group Members, 2024). These databases cover a wide range of data, including protein information in UniProt, gene expression in Bgee, enzymatic reactions in Rhea, protein interactions in STRING, cell lines in Cellosaurus, and orthologs in OMA and OrthoDB. All these resources are listed on Expasy, the Swiss Bioinformatics Resource Portal.

Life science data is inherently diverse, differing in format, storage, and access methods, which makes data integration and reuse particularly challenging. To overcome this, SIB has contributed to building a "semantic web" of biological data, featuring public SPARQL endpoints including Bgee, Cellosaurus, GlyConnect, STRING, UniProt (of Bioinformatics RDF Group Members, 2024).

SPARQL endpoints are specialized websites that can process so-called SPARQL queries, enabling users to perform searches that go beyond simple text-based queries such as "Find UniProt entries with a transmembrane region, with an Alanine in the 15 amino acid region preceding the transmembrane" (Bolleman et al., 2025).

Additionally, SPARQL allows users to execute federated queries, retrieving data from multiple distributed sources simultaneously such as "Identify mouse homologs in the OMA Browser for human enzymes that are involved in sterol-related reactions, as described in the Rhea database" (Bolleman et al., 2025; Moretti et al., 2026). However, constructing SPARQL queries is often complex and requires expertise that many users do not have.

### 4.1 HOW LARGE LANGUAGE MODELS HELP

Advances in Large Language Models (LLMs), like ChatGPT, have opened new possibilities in natural language processing. These models can be harnessed to translate user questions directly into structured queries, enabling users to interact with data in plain language without needing to understand complex syntax or underlying knowledge graph structure (O'Neil et al., 2024; Madani et al., 2023).

## 5 EXPASYGPT: A GATEWAY TO FEDERATED KNOWLEDGE GRAPHS

ExpasyGPT harnesses the power of LLMs to translate user questions directly into SPARQL queries, enabling users to interact with data in plain language without needing to understand the complex SPARQL syntax or underlying knowledge graph structure (Bolleman et al., 2025).

The SIB SPARQL endpoints are developed and maintained by various research groups. Over the years, these groups have worked to enrich and harmonize endpoint documentation and metadata (of Bioinformatics RDF Group Members, 2024). ExpasyGPT builds on a large language model (LLM) using Retrieval-Augmented Generation (RAG), to incorporate the harmonized metadata and 1000 example questions—including 65 federated queries (Bolleman et al., 2025). These examples are publicly available on GitHub and are linked to their respective SPARQL endpoints (e.g., UniProt SPARQL examples). In addition to improving the LLM's ability to generate accurate and contextual responses, these examples also support non-expert users by guiding them in constructing their own SPARQL queries.

### 5.1 HOW EXPASYGPT WORKS:

ExpasyGPT is part of a larger effort to enhance search capabilities across SIB's interoperable databases, making life science data more accessible to everyone. By connecting an RAG-enhanced LLM to a set of SPARQL endpoints, ExpasyGPT enables access to structured biological knowledge in natural language. The currently supported endpoints include:

- UniProt, an expertly curated database of proteins
- OMA, the orthologous matrix
- Bgee, an expertly curated gene expression database

- Rhea, an expertly curated database of biochemical reactions
- Cellosaurus, an expertly curated database on cell lines
- SwissLipids, an expertly curated database of lipids.

As the service evolves, it will integrate additional endpoints from the SIB Knowledge Graph (of Bioinformatics RDF Group Members, 2024).

## 6 STRING-CHAT: CONVERSATIONAL INTERFACE FOR PROTEIN INTERACTION NETWORKS

The STRING database is a core resource for predicting and analyzing protein-protein interactions, systematically collects and integrates protein–protein interactions—both physical interactions as well as functional associations. The data originate from a number of sources: automated text mining of the scientific literature, computational interaction predictions from co-expression, conserved genomic context, databases of interaction experiments and known complexes/pathways from curated sources. All of these interactions are critically assessed, scored, and subsequently automatically transferred to less well-studied organisms using hierarchical orthology information. The data can be accessed via the website, but also programmatically and via bulk downloads (Szklarczyk et al., 2023). Like the SIB resources, STRING provides a Resource Description Framework (RDF) data model and SPARQL endpoint, enabling to access protein-protein interaction data without using the traditional web page interface. This SPARQL endpoint allows advanced querying of its vast interaction network, which integrates data from genomic context predictions, high-throughput experiments, co-expression, and automated text-mining. One of the key benefits of the SPARQL endpoint, over the API, is its capacity to handle "federated queries". These federated queries allow for simultaneous querying and integration across multiple databases, eliminating the need for centralization and crafting individual parsers for each of the resource APIs. For instance, with a single query, can amalgamate data about protein features from UniProt with data about interaction partners from STRING (spa).

However, crafting SPARQL queries to traverse protein networks, identify interacting proteins, or retrieve functional annotations remains a technical hurdle. STRING Chat addresses this by providing a natural language interface powered by an LLM. Users can ask questions in plain English about proteins, their functions, interactions, and associated diseases.

### 6.1 HOW STRING-CHAT WORKS:

STRING Chat is a natural-language interface that is built on a Model Context Protocol (MCP) server. It serves as a working example of how AI agents can access and combine STRING data in real time using structured tool definitions exposed by the MCP (str, b).

It supports natural language queries such as:

- "Show me the interaction network for TP53."
- "What are the homologs of BRCA1 in zebrafish?"
- "Which pathways are enriched in this list of proteins?"

### 6.2 USE CASES:

STRING Chat is best suited for quick exploration, conversational queries. It can invoke multiple MCP tools in sequence, aggregate their outputs, and summarize the combined results into a unified response. The interface supports all major languages, and example questions and use cases are shown directly below the chat input (str, b).

#### 6.2.1 EXAMPLE PROMPTS AND CAPABILITIES (STR, A):

- **Network Exploration:** "Show me a network of INSR, IRS1, and AKT1, including their interaction neighborhood." The chat generates a query from the STRING data and invokes the MCP tools to fetch these proteins and their known interactors, then renders a visual

network where nodes are proteins and edges are interactions, colored by evidence type. (Fig. 1).

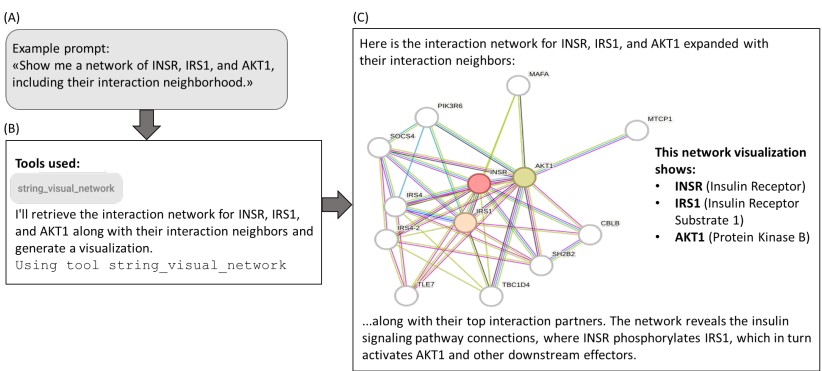

Figure 1: An example of STRING-Chat operation. The key stages are shown, starting from the query and ending with the system's response: A) Query example: "Show me a network of INSR, IRS1, and AKT1, including their interaction neighborhood.", B) Using the string_visual_network tool to visualize a network for INSR, IRS1, and AKT1, C) Visualized interaction network for INSR, IRS1, and AKT1.

- **Homology Search:** "What are the homologs of the fly's 'ci' protein in vertebrates?". The tool queries orthology data and lists the homologous proteins. (Fig. 2).

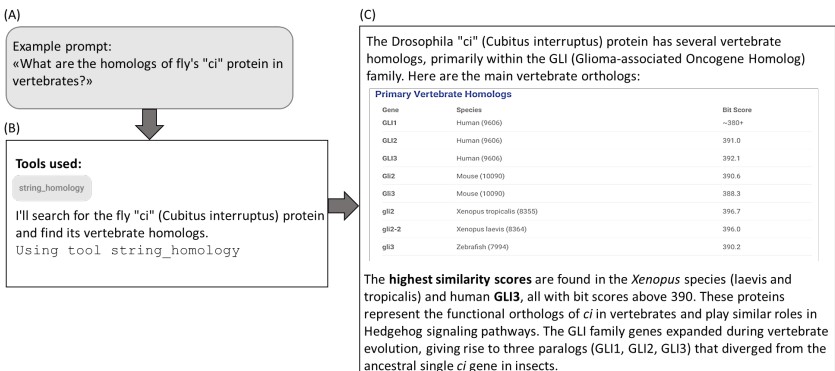

Figure 2: An example of STRING-Chat operation. The key stages are shown, starting from the query and ending with the system's response: A) Query example: "What are the homologs of fly's "ci" protein in vertebrates?", B) Using the string_homology tool to search for homologs of the Drosophila ci protein, C) Resulting list of vertebrate homologs from the GLI family with similarity scores and evolutionary interpretation.

- **Disease Association:** "Show clustered network of Type I diabetes associated proteins." The chat generates a query from the STRING data and invokes the MCP tools — to retrieve proteins associated with Type I diabetes (from the KEGG pathway) and to organize them into functional modules — presenting a clustered interaction network of proteins. (Fig. 3).

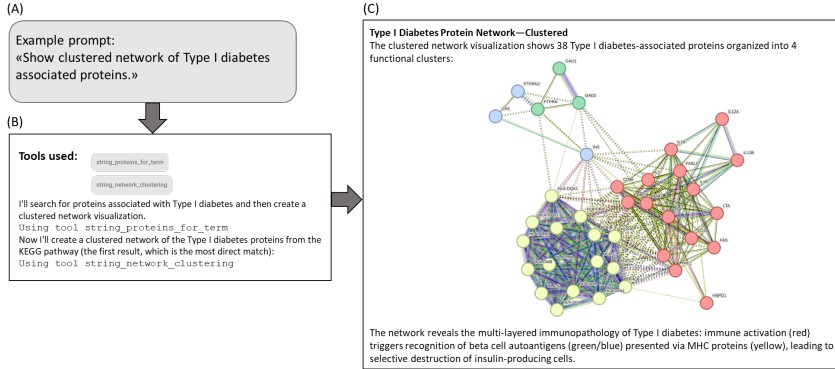

Figure 3: An example of STRING-Chat operation. The key stages are shown, starting from the query and ending with the system's response: A) Query example: "Show clustered network of Type I diabetes associated proteins.", B) Using the string_proteins_for_term tool to retrieve Type I diabetes-associated proteins (from the KEGG pathway) and the string_network_clustering tool to organize them into functional modules, C) Visualized clustered interaction network for Type I diabetes-associated proteins showing 38 proteins organized into 4 functional clusters: immune activation and cytokine signaling (red), MHC/HLA complex for antigen presentation (yellow), beta cell autoantigen targets (green), and insulin secretion and beta cell function (blue).

# 7 MEGA 12.1 AND MEGA-GPT: IMPROVING PHYLOGENETIC ANALYSIS THROUGH SOFTWARE IMPROVEMENTS AND ARTIFICIAL INTELLIGENCE-DRIVEN MANAGEMENT

MEGA (Molecular Evolutionary Genetics Analysis) software is widely used for molecular evolutionary and phylogenetic analyses. Its latest cross-platform release, MEGA 12.1, extends native functionality to macOS and modern Linux distributions while retaining all methodological and performance enhancements of the Windows version. Key improvements include:

1. Accelerated Maximum Likelihood (ML) analyses via a filtered model test, adaptive bootstrapping, and fine-grained parallelization;

2. A modernized graphical user interface with enhanced visualization and support for high-resolution displays;

3. An improved Calibration Editor integrated with the TimeTree database to facilitate molecular dating;

4. Full cross-platform session file compatibility, enabling seamless collaboration across operating systems. These updates collectively improve the accessibility, computational efficiency, and user experience of MEGA in diverse research environments (Stecher et al., 2025).

MEGA-GPT, an AI-driven resource that leverages ChatGPT augmented with retrieval techniques to guide users through MEGA's analytical workflows via natural language queries. By integrating MEGA's help documentation, version-specific articles, and other key publications, MEGA-GPT enhances ChatGPT's standard responses to deliver step-by-step protocols, clarify analytical settings, and recommend optimal workflows (Allard and Kumar, 2025).

## 7.1 MEGA-GPT: RAG-POWERED WORKFLOW TRANSLATION.

MEGA-GPT is built on a retrieval-augmented generation (RAG) framework, which combines dynamic information retrieval with language generation. It takes a hybrid approach, which is particularly effective for domain-specific applications, as it enables large language models to access up-to-date, specialized information beyond their static training data, thereby mitigating common issues like factual hallucination and domain insensitivity related to using artificial intelligence.

The RAG architecture comprises two key components: a retriever and a generator. In the context of MEGA- GPT:

1. The Retriever scans an indexed knowledge base containing the software's comprehensive documentation, established analysis protocols, and solutions to common user tasks.

2. The Generator then synthesizes a coherent and actionable response by integrating the retrieved, authoritative technical information with the user's original query.

This two-step process grounds the AI's guidance in the specific, structured workflow logic of the MEGA software, enhancing factual accuracy and reducing errors. Crucially, the RAG framework allows for continuous updates to the knowledge base as MEGA evolves—with new methods, features, or documentation—without requiring extensive and costly retraining of the core LLM. This adaptability is crucial in rapidly evolving scientific fields and for software under continuous development, ensuring that MEGA-GPT delivers guidance that reflects the latest methodologies included in MEGA (Allard and Kumar, 2025).

### 7.1.1 EXAMPLE PROMPTS AND FUNCTIONALITY (MEG):

- **Methodological guidance and workflow planning:** "I want to create a Maximum Likelihood tree". The chat explains how to prepare sequences (e.g., "use MAFFT or Clustal Omega"), recommends a model selection program (e.g., jModelTest), describes the logic of tree construction, and provides an example command line for RAxML. It also explains what bootstrapping is and how to run it (e.g., by adding the -# 100 flag in RAxML) and recommends a program for visualization (e.g., FigTree or iTOL).

- **Conceptual education and tool selection assistance:** "How can I align protein sequences?". The chat explains the difference between global and local alignment, mentions core algorithms (e.g., Needleman-Wunsch, Smith-Waterman) and their purposes, and then recommends and provides a detailed, step-by-step guide for performing alignment directly in the MEGA 12.1 graphical interface. It explains how to choose the correct algorithms in MEGA (e.g., MUSCLE or Clustal Omega) and offers general advice for obtaining high-quality protein alignments within the capabilities of the program.

- **Construction of simple phylogenetic trees using text-based graphics:** "Give an example and build a simple phylogenetic tree." The chat creates a minimal yet biologically meaningful set of sequences (e.g., Human, Chimp, Mouse, Frog), provides a step-by-step calculation of pairwise distances and interprets them (e.g., "Human and Chimp are the closest"), presents a diagram of the expected tree in the form of text-based graphics (ASCII art), clearly reflecting the logic of evolutionary relatedness, and provides steps to build the tree in the MEGA 12.1 graphical interface, starting from data entry (Align → Edit/Build Alignment) and ending with the selection of the phylogenetic inference method and model (Phylogeny → Construct/Test Neighbor-Joining Tree). (Fig. 4).

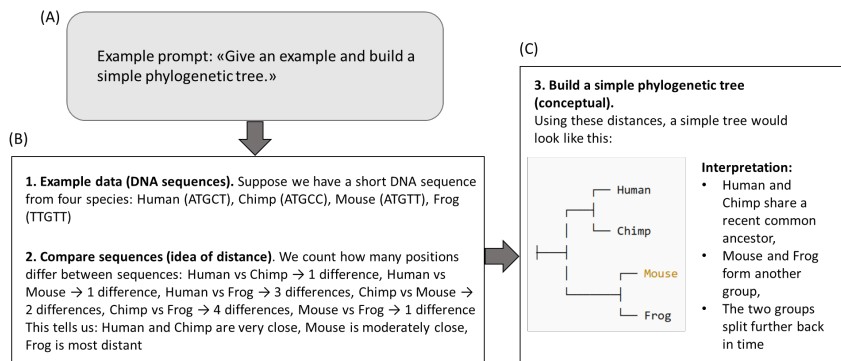

Figure 4: An example of MEGA-GPT operation. The key stages are shown, starting from the query and ending with the system's response: A) Query example: "Give an example and build a simple phylogenetic tree." B) Data example (proposed short DNA sequences of four species: Human (ATGCT), Chimp (ATGCC), Mouse (ATGTT), Frog (TTGTT)) and the calculation of pairwise genetic distances with their interpretation. C) Visualization of the conceptual tree topology (ASCII art) and its biological interpretation.

Thus, while chat interfaces such as ExpasyGPT or STRING-chat act as intelligent "translators, "converting natural language into database queries (SPARQL) and returning ready-made results, MEGA-GPT performs a different task: providing detailed instructions for executing workflows in the field of bioinformatics. Its essence lies not in automatically executing queries, but in its ability to guide the user through a complex, multi-step workflow within specialized software that supports the loading and analysis of user files. This makes it not a substitute, but a powerful context-dependent addition to general language models for solving specific tasks of phylogenetic analysis.

## 8 SEMANTIC WEB AND KNOWLEDGE GRAPHS: KEY CONCEPTS

**RDF and Knowledge Graphs.** The Semantic web provides a common framework that allows data to be shared and reused across application, enterprise, and community boundaries (of Bioinformatics RDF Group Members, 2024). It is based on the Resource Description Framework (RDF), a graph data model. Resource Description Framework (RDF) is a standard semantic graph technology suited to sharing and linking data worldwide. RDF triples consist of a subject, predicate and object. The predicate specifies the relationship between the subject and object, each defined by a globally unique identifier. Triples can thus be represented as a graph, where the subject and object correspond to nodes, and the predicate the edge joining the nodes. By connecting all the information about entities found in RDF triples, it is possible to construct a Knowledge Graph that represents information about entities (e.g. proteins, genes, organs) and their relationships to one another (e.g. 'is expressed in', 'codes for').

**Querying the Knowledge Graph with SPARQL.** A SPARQL endpoint enables users (human or machines) to query the RDF data using SPARQL. The SIB databases which provide a SPARQL endpoint are listed at: `https://www.expasy.org/search/sparql` (of Bioinformatics RDF Group Members, 2024). SPARQL (SPARQL Protocol and RDF Query Language) is a query language for retrieving and manipulating data stored in Resource Description Framework (RDF) format. SPARQL allows search criteria for specific content to be combined, allowing the user to perform queries which cannot be answered with text-based search. It thus provides a means to mine the information stored in databases. Furthermore, SPARQL enables data distributed across multiple sources to be queried by executing federated queries.

A federated query is a special query that runs on more than one SPARQL endpoint, enabling cross-database querying and information retrieval. Although federated queries require knowledge of the data models of the databases to be queried, and of which entities are equivalent, they are nevertheless extremely powerful, allowing users to explore the data in databases worldwide, provided they have SPARQL endpoints. A set of SPARQL examples that use the different SIB resources is found at: `https://sib-swiss.github.io/sparql-examples/` (Bolleman et al., 2025).

**Helping users query the knowledge graph.** Large Language Models (LLMs) are creating a shift of paradigm in how we interact with data across domains. Bioinformatics is one of the fields most prominently impacted by the advent of LLMs, whether for biodata exploration, via LLM-based AI assistants or for dedicated, domain-specific LLMs such as Protein Language Models. While LLMs are prone to mistakes in factual recall, their ability to summarize and to use tools suggest new opportunities to help non-expert users query and interact with complex data, while drawing on the Knowledge Graph to improve reliability of the answers (of Bioinformatics RDF Group Members, 2024).

Writing SPARQL queries is still beyond the expertise of most users. Given the current progress in LLMs and the demonstrated importance of documentation for accessing knowledge graphs with tools like ChatGPT, the potential of these models to generate federated queries in response to user questions is being explored (Bolleman et al., 2025; O'Neil et al., 2024). In doing so, the current search capabilities of Expasy should be significantly enhanced and moving towards a unified search engine across the interoperable SIB knowledge graphs (of Bioinformatics RDF Group Members, 2024).

## 9 GENE NETWORKS AND AI TOOLS: RESEARCH EXAMPLES

Recent studies have demonstrated the application of gene network reconstruction (STRING-DB and GeneMANIA) and AI tools for studying molecular mechanisms, diseases, and biological systems.

In plant biology, several studies have employed network-based approaches. Antropova et al. (2024) performed computational identification of genetic markers associated with molecular mechanisms of reduced rice resistance to Rhizoctonia solani under excess nitrogen fertilization. Kleshchev et al. (2024) reconstructed and analyzed the microRNA regulation gene network in wheat drought response mechanisms. Samarina et al. (2024) used transcriptomic approaches to reveal core pathways of nitrogen deficiency response in tea plants (*Camellia sinensis* L.). Eftekhari et al. (2024) presented a study on the application of artificial intelligence, machine learning, and deep learning in plant breeding.

In human disease research, the glioblastoma gene network was reconstructed and ontology analysis was performed using online bioinformatics tools, identifying key genes and pathways involved in the pathogenesis of glioblastoma (Gubanova et al., 2021). In a study on schizophrenia (Dokhoyan et al., 2022), the schizophrenia gene network was reconstructed to identify potential target genes for therapeutic intervention, using database searches and network analysis to characterize the molecular mechanisms underlying the disorder. In a study on Angelman syndrome, a neurogenetic disorder (Karpyn et al., 2024), a computer reconstruction of gene interactions associated with this condition was performed, mapping the relationships between known disease-associated genes. Other studies have applied similar approaches to Parkinson's disease (Orlov et al., 2021), diffuse large B-cell lymphoma (Voropaeva et al., 2025), and glioma (Turkina et al., 2023).

In animal models, Chadaeva et al. (2024) analyzed differentially expressed genes in brain regions of rats selected for tame or aggressive behavior and used STRING-DB to construct gene association networks, identifying hub genes involved in behavioral regulation.

Several bioinformatics tools have expanded the capabilities of gene network analysis and AI-driven prediction. Aristarkhov et al. (2023) developed a neural network-based web-service for microRNA target prediction, demonstrating deep learning applications in gene regulation analysis. Tiis et al. (2021) employed STRING-DB to reconstruct xenobiotic metabolism gene networks, identifying key interactions between NAT2 and cytochrome P450 genes. The ANDSystem platform, particularly its ANDDigest module (Ivanisenko et al., 2020), provides AI-driven text-mining capabilities comparable to STRING and GeneMANIA, enabling knowledge extraction from PubMed and gene network reconstruction with integrated trend analysis.

The development of AI approaches is linked to their integration into medical practice. Sechenov University develops AI applications for digital health, including risk sharing agreements for pharmacotherapy risk management (Oborotov et al., 2023) and holistic approaches for AI implementation in pharmaceutical products lifecycle (Koshechkin et al., 2022). These developments open up prospects for transitioning from fundamental research to real-world clinical practice.

## 10 CONCLUSION

This review demonstrates the revolutionary impact of large language models (LLMs) on the development of bioinformatics and systems biology. Modern artificial intelligence-based tools, such as ExpasyGPT, STRING-chat, MEGA-GPT, scGPT, and GP-GPT, are fundamentally changing the approach to biological data analysis, making it more accessible and efficient for researchers with varying levels of expertise.

The main achievement has been the creation of intuitive interfaces capable of transforming user natural language into structured database queries. This significantly simplifies work with biological data and allows researchers to focus on analyzing results rather than on the technical aspects of data handling.

An important area of development is the creation of specialized tools for specific bioinformatics domains, applied in fields such as gene expression analysis, evolutionary biology, and structural genomics. The integration of LLMs with existing databases, including KEGG, PathBank, and Reactome, provides a powerful foundation for biomedical research and disease modeling.

A promising direction for future development remains the integration of various tools into a unified system, despite existing challenges related to data unification and ontology alignment. Particular attention is being paid to improving the accuracy, reproducibility, and accessibility of results generated by AI systems.

The implementation of LLMs in bioinformatics opens new horizons for research in protein structure prediction, drug development and interaction analysis, and therapeutic target discovery, making these tools indispensable assistants for modern researchers.

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
