# OpenReview forum: "ChatGPT like tools for gene networks reconstruction"
_mathai.club/MathAI/2026/Conference — MathAI 2026 Conference Submission_

### Official Review · Reviewer_8S53 · 2026-03-11
**Despite the relevance of the topic LLMs in bioinformatics, the paper does not present new structures, datasets, or methodologies, it provides a descriptive overview of existing approaches.**

**Rating:** 3
**Confidence:** 4

**Review:**

Quality
The work is predominantly descriptive. While it lists many tools, it does not critically evaluate them against each other.
Originality
Despite the relevance of the topic, the paper does not present new structures, datasets, or methodologies, it provides a descriptive overview of existing approaches.
Significance
The integration of large language models into bioinformatics is a rapidly developing field. Therefore, it is necessary to summarize the existing research results, but their critical analysis is necessary.
Сlarity
There are disadvantages in text formatting and structure. The numbering of paragraphs and sections in some cases looks redundant and not entirely logical.

---

### Official Review · Reviewer_PkV2 · 2026-03-12
**The manuscript addresses a timely topic but lacks academic presentation**

**Rating:** 3
**Confidence:** 4

**Review:**

The paper does not meet the standard of a conference publication.

Strengths

The manuscript also highlights an important practical point: natural-language interfaces may lower the barrier for researchers who are not familiar with SPARQL, semantic web technologies, or specialized database query languages. Some of the discussed tools and examples may be useful for readers who are new to this area.

Weaknesses

The main weakness is the lack of a clear scientific contribution. The paper is presented as a review, but it does not define a review methodology, inclusion criteria, comparison framework, or taxonomy of the considered systems. As a result, it remains a loose narrative overview rather than a rigorous survey or analytical study. The title suggests a focused discussion of gene network reconstruction, but much of the content is devoted to broader topics such as semantic web infrastructure, drug discovery, and phylogenetic workflow assistance, which makes the scope diffuse.